# Understanding Few-Shot Learning: Measuring Task Relatedness and Adaptation Difficulty via Attributes

**Minyang Hu**[1,2]**, Hong Chang**[1,2]**, Zong Guo**[1,2]**, Bingpeng Ma**[2]**, Shiguang Shan**[1,2]**, Xilin Chen**[1,2]

[1] Institute of Computing Technology, Chinese Academy of Sciences
[2] University of Chinese Academy of Sciences
{minyang.hu, zong.guo}@vipl.ict.ac.cn, {changhong, sgshan, xlchen}@ict.ac.cn, bpma@ucas.ac.cn

## Abstract

Few-shot learning (FSL) aims to learn novel tasks with very few labeled samples by leveraging experience from *related* training tasks. In this paper, we try to understand FSL by exploring two key questions: (1) How to quantify the relationship between *training* and *novel* tasks? (2) How does the relationship affect the *adaptation difficulty* on novel tasks for different models? To answer the first question, we propose Task Attribute Distance (TAD) as a metric to quantify the task relatedness via attributes. Unlike other metrics, TAD is independent of models, making it applicable to different FSL models. To address the second question, we utilize TAD metric to establish a theoretical connection between task relatedness and task adaptation difficulty. By deriving the generalization error bound on a novel task, we discover how TAD measures the adaptation difficulty on novel tasks for different models. To validate our theoretical results, we conduct experiments on three benchmarks. Our experimental results confirm that TAD metric effectively quantifies the task relatedness and reflects the adaptation difficulty on novel tasks for various FSL methods, even if some of them do not learn attributes explicitly or human-annotated attributes are not provided. Our code is available at https://github.com/hu-my/TaskAttributeDistance.

## 1 Introduction

Learning in human biological system is highly efficient compared to artificial systems. For example, only one example is enough for a child to learn a novel word [9], while thousands of training samples are needed for deep learning models. This learning efficiency comes from the past experiences accumulated by human brain. Inspired by human learning capability, Few-Shot Learning (FSL) aims to learn novel tasks with very few samples by leveraging experience from *related* training tasks.

Despite the great progress made in FSL during the past decade [48, 42, 17, 45, 22, 10, 16, 50, 27], there is less exploration on what it means for training tasks to be related. Some prior studies [48, 17, 45] focus on constructing related training tasks by sampling categories within a dataset. Recent works [46, 33] consider training tasks sampled from different datasets, known as cross-domain few-shot learning. The latter is considered as a more challenging problem, partly because the category gaps across datasets are usually larger, resulting in training tasks less related to novel tasks. Although this follows the intuition, two natural questions arise: (1) How to quantify the relationship between *training* and *novel* tasks? (2) How does the relationship affect the *adaptation difficulty* on novel tasks for different FSL models?

To quantify the relationship between tasks, one approach is to measure the category relationship in a common representation space. With properly learned representation function, various distribution divergence metrics, like EMD [39] and FID [20], can be used to calculate the distance between feature distributions of different categories, as well as tasks over them. The main obstacle to this

solution is how to learn a good representation function for novel categories with only a few labeled samples. Previous works [32, 33] assume that the representation function learned from training categories can be directly applied to novel categories, but this assumption may not hold due to the large category gaps. Some recent studies [1, 26] utilize the Fisher Information Matrix to quantify the task relatedness without assuming a common representation function. However, this approach has a heavy burden of computing Hessian matrix. Besides, the calculated task distances are highly dependent on the learned model, making them difficult to apply to other FSL models.

To investigate the influence of task relatedness on the difficulty of adapting models to novel tasks, existing studies [37, 40, 32, 33] often make an empirical assumption that a novel task with large distances to the training tasks will be hard to adapt to. With this assumption, some works [40, 32, 33] employ specific distance metrics to quantify the task relatedness, and then construct more challenging benchmarks to explore the generalization ability of different models. Different with the these works, [26] select the most related training data to improve the episodic fine-tune process for FSL models based on an asymmetric task relatedness measure. Despite their empirical success, all the aforementioned works lack theoretical analysis, leaving the connection between task relatedness and task adaptation difficulty not formally clarified.

In this work, we try to overcome the existing obstacles and answer the two questions via attributes. Firstly, we introduce Task Attribute Distance (TAD) as a metric to quantify the task relatedness. Our intuition lies in the attribute compositionality hypothesis (shown in Fig. 1): each category can be represented as a composition of attributes, which are reusable in a huge assortment of meaningful compositions. TAD formulates this intuition by first measuring the total variation distance between attribute distributions of training and novel categories, and then finding the maximum matching with minimum weight of a bipartite graph. Unlike other metrics, TAD only relies on the attribute conditional distributions, making it independent of models and the number of samples in novel tasks. Secondly, we utilize the TAD metric to establish a theoretical connection

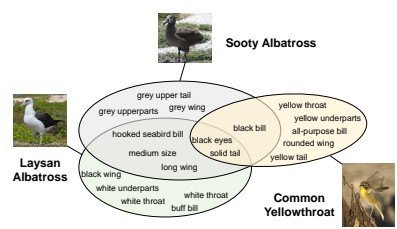

Figure 1: The categories and attribute sets of three bird images. Each category can be represented as a composition of some attributes, which act as a relationship bridge between different categories.

between task relatedness and task adaptation difficulty in FSL. We provide theoretical proof to verify that TAD contributes to the *generalization error bound on a novel task*, at different training settings. Importantly, our theoretical results indicate that the adaptation difficulty on a novel task can be efficiently measured by relying solely on the TAD metric, without the need for training any models.

We conduct experiments on three benchmark datasets to validate our theoretical results and explore the generality of TAD metric. To this end, we consider two scenarios that the human-annotated attributes are provided or not available. For the latter case, we propose an efficient way to auto-annotate attributes. Experimental results show that as the distance between novel and training tasks increases, the performance of various FSL models on novel tasks decreases. With the TAD metric, we also present two potential applications: intervening a small set of difficult novel tasks before training, and investigating the benefits of heterogeneous training tasks for different models.

Our main contributions can be summarized as follows:

- We propose Task Attribute Distance (TAD) metric in FSL to quantify the relationship between training and novel tasks, which is model-agnostic and easy to compute.

- We provide a theoretical proof of the generalization error bound on a novel task based on TAD, which connects task relatedness and adaptation difficulty theoretically.

- We conduct experiments to show TAD can effectively reflect the adaptation difficulty on novel tasks for various FSL methods with human-annotated and auto-annotated attributes.

## 2 Related works

**Few-shot learning.** Most recent few-shot learning (FSL) approaches are based on meta-learning framework. These meta-learning based methods can be roughly categorized into two categories. The

metric-based approach [48, 42, 45, 22, 3, 16, 2, 50] aims to learn a cross-task embedding function and predict the query labels based on the learned distances. The optimization-based approach focuses on learning some optimization state, like model initialization [17, 44] or step sizes [28, 5], to rapidly update models with very few labeled samples. Except for these meta-learning methods, transfer-based approaches [10, 11] have recently shown that standard transfer learning procedure of early pre-training and subsequent fine-tuning is a strong baseline for FSL with deep backbone networks.

**Meta-learning theory.** In parallel to the empirical success of meta-learning methods, a series of theoretical works study how meta-learning utilizes the knowledge obtained from the training task data and generalizes to the novel test tasks. Many works [35, 4, 38, 12, 19] give a generalization error bound on novel tasks from the PAC-Bayesian perspective [30, 31]. These works often assume that each task is sampled from a meta task distribution. Under such assumption, the generalization error bound on novel tasks can be reduced with increasing number of training tasks and training samples [4]. Some recent works replace the meta task distribution assumption with other conditions. For example, [8, 14, 47] assume a common representation function between different tasks, then they give the sample complexity bounds for the expected excess risk on a novel task. However, the above works do not quantify the relationship between training and novel tasks, and seldom explore the adaptation difficulty on novel tasks. In this paper, we propose the TAD as a metric to quantify the task relatedness, and provide a new generalization error bound on a novel task.

**Task Difficulty.** Exploring the difficulty of few-shot tasks is an important research question in FSL, and has been explored from two aspects: measure the difficulty of (1) training tasks and (2) novel tasks. For training tasks, many previous works [44, 29, 43, 6] have attempted to measure their difficulties based on a model's output, such as negative log-likelihood or accuracy, and use this information to sample different tasks to train the model. The sampling strategy based on training task difficulty is similar to the area of hard example mining [41] or curriculum learning [7] that trains a model according to a specific order of samples to improve its generalization ability. For novel tasks, recent works [1, 37, 32, 40, 26, 33] try to measure the adaptation difficulty of novel tasks based on the relationship between novel and training tasks. Task2vec [1] encodes each novel task into an embedding based on the Fisher Information Matrix, and uses the norm of embedding to reflect task difficulty. Other works utilize specific distribution divergence metrics, such as EMD and FID, to quantify the task relatedness, then select the most related training data to fine-tune models [26] or construct more challenging benchmarks to explore the generalization of models [37, 40, 33]. Closer to our work, [37] introduces a transferability score to quantify the transferability of novel tasks via attributes and explore the generalization of FSL models at different transferability scores. However, different with our work, [37] aims to investigate the benefits of self-supervised pretraining with supervised finetuning in the few-shot attribute classification context.

## 3 Problem Setting

In few-shot learning problem, a model observes $n$ different training tasks $\{\tau_i\}_{i=1}^n$ with each task represented as a pair $\tau_i = (\mathcal{D}_i, S_i), 1 \leq i \leq n$. $\mathcal{D}_i$ is an unknown data distribution over the input space $\mathcal{X}$ and label space $\mathcal{Y}_i$. $S_i = \{(x_k, y_k)\}_{k=1}^{m_i}, (x_k, y_k) \sim \mathcal{D}_i$ represents an observed training set drawn from $\mathcal{D}_i$. With a model trained on the $n$ training tasks, our target is to adapt and evaluate it on $t$ novel test tasks $\tau_j' = (\mathcal{D}_j', S_j'), 1 \leq j \leq t$. $\mathcal{D}_j'$ is an unknown data distribution over $\mathcal{X}$ and $\mathcal{Y}_j'$, and $S_j'$ is a labeled dataset drawn from $\mathcal{D}_j'$. Note that, in few-shot learning, the labeled data for each category is very limited, and the categories in training tasks $\{\tau_i\}_{i=1}^n$ will not appear in novel tasks $\{\tau_j'\}_{j=1}^t$, which means $\mathcal{Y}_i \cap \mathcal{Y}_j' = \emptyset, \forall i \in \{1, ..., n\}, j \in \{1, ..., t\}$.

## 4 Generalization Analysis of Attribute-based Few-shot Learning

The key to answering the two questions we raised above lies in a proper metric that quantifies task relatedness and contributes to the generalization error on a novel task. In this section, we begin by introducing the Task Attribute Distance (TAD), which first measures the category relationship through attribute conditional distributions, based on which the relationship between tasks is measured as well. Then, to facilitate our theoretical analysis, we introduce a meta-learning framework with attribute learning. Finally, we provide theoretical proof that establishes a connection between the proposed TAD metric and the generalization error bound on a novel task.

### 4.1 Model-Agnostic Distance Metric via Attributes

To measure the relationship between two categories $y_k$ and $y_t$, a natural idea is to measure the divergence between class-conditional distributions $p(x|y_k)$ and $p(x|y_t)$ by a divergence measure, like the $L_1$ or total variation distance (TV) [18]. Unfortunately, this idea is frequently unfeasible, because the set of measurable subsets under distributions $p(x|y_k)$ and $p(x|y_t)$ is often infinite, making it hard to compute the distance from finite samples.

In this paper, we introduce attributes to overcome the above limitation because the values of attributes are usually discrete and finite. The basic idea is based on the *attribute compositionality hypothesis* presented in the introduction section. Specifically, the distance between two categories can be measured by the difference between the corresponding attribute conditional distributions. Follow the above idea, let $\mathcal{A}$ be the attribute space spanned by $L$ attribute variables $\{a^l\}_{l=1}^L$, we define the *distance between two categories* $y_k$ and $y_t$ on $L$ attribute variables as:

$$d(y_k, y_t) = \frac{1}{L} \sum_{l=1}^L d_{TV}(p(a^l|y_k), p(a^l|y_t)) = \frac{1}{L} \sum_{l=1}^L \sup_{B \in \mathcal{B}^l} \left| P_{a^l|y_k}[B] - P_{a^l|y_t}[B] \right|, \quad (1)$$

where $\mathcal{B}^l$ is the set of measurable subsets under attribute conditional distributions $p(a^l|y_k)$ and $p(a^l|y_t)$, and $\mathcal{B}^l$ is a finite set if the values of variable $a^l$ are finite.

Based on the distance between categories, we then define the distance metric between training and novel test tasks. To this end, we represent the set of categories in training task $\tau_i$ and the set of categories in novel task $\tau_j'$ as two disjoint vertex sets respectively, and construct a weighted bipartite graph $G$ between the two vertex sets, where each node denotes a category and each edge weight represents the distance between two categories in $\tau_i, \tau_j'$ respectively. Let $M = \{e_{kt}\}$ denote a maximum matching of $G$, which contains the largest possible number of edges and any two edges do not connect the same vertex. We choose $M$ with minimum weights and define the *task distance* as

$$d(\tau_i, \tau_j') = \frac{1}{|M|} \sum_{e_{kt} \in M} d(y_k, y_t) = \frac{1}{L|M|} \sum_{e_{kt} \in M} \sum_{l=1}^L d_{TV}(p(a^l|y_k), p(a^l|y_t)), \quad (2)$$

where $|M|$ is the number of edges in matching $M$. From this definition, if the two tasks $\tau_i, \tau_j'$ are identical, the task distance $d(\tau_i, \tau_j')$ is equal to 0. We call the task distance $d(\tau_i, \tau_j')$ as the Task Attribute Distance (TAD). Additionally, the TAD is invariant to the permutation of categories in tasks. In other words, modifying the numeric order of categories in $\tau_i$ or $\tau_j'$ does not affect their distance $d(\tau_i, \tau_j')$. To emphasize the attribute conditional distributions with respect to different tasks, we hereinafter add task indexes on the distribution and probability notations, like $p^i(a^l|y_k)$ and $P_{a^l|y_k}^i[B]$, although the true distributions are task agnostic.

### 4.2 A Specific Few-Shot Learning Framework

To facilitate the subsequent theoretical analysis, we consider a meta-learning framework with attribute learning. In this framework, a meta-learner $f_\theta : \mathcal{X} \to \mathcal{A}$ parameterized by $\theta$ learns the mapping from a sample $x \in \mathcal{X}$ to attributes $a \in \mathcal{A}$, and a task-specific base-learner $g_{\phi_i} : \mathcal{A} \to \mathcal{Y}_i$ parameterized by $\phi_i$ learns the mapping from attributes $a \in \mathcal{A}$ to a class label $y \in \mathcal{Y}_i$ for training task $\tau_i$. During training, $f_\theta$ is meta-learned from $n$ different training tasks. To adapt to a novel task $\tau_j'$, we fix $f_\theta$ and train a new base-learner $g_{\phi_j'}$ parameterized by $\phi_j'$ using few labeled samples $S_j'$.

In the following theoretical deduction, we will denote $p_\theta(a, y) \triangleq p(f_\theta(x), y)$ as a joint distribution over $\mathcal{A} \times \mathcal{Y}$ induced by the meta-learned mapping $f_\theta$. Further, with the induced conditional distribution $p_\theta(a|y)$, we can compute the distance between task $\tau_i$ and $\tau_j'$ as

$$d_\theta(\tau_i, \tau_j') = \frac{1}{L|M|} \sum_{e_{kt} \in M} \sum_{l=1}^L d_{TV}(p_\theta^i(a^l|y_k), p_\theta^j(a^l|y_t)). \quad (3)$$

Note that $d(\tau_i, \tau_j')$ (as defined in Eq. (2)) computed based on $p(a|y)$ is the model-agnostic distance metric, while $d_\theta(\tau_i, \tau_j')$ is a model-related one since it relies on the meta-learned mapping $f_\theta$.

## 4.3 Theoretical Analysis on Generalization

As for the above meta-learning framework, we provide theoretical analysis of the generalization error on a novel task, in terms of the proposed TAD metric. We define the generalization error and empirical error of the meta-learner $f_\theta$ on novel task $\tau'_j$ as

$$\epsilon(f_\theta, \tau'_j) = \mathbb{E}_{(x,y)\sim\mathcal{D}'_j}[\mathbb{I}(g_{\phi'_j}(f_\theta(x)) \neq y)], \quad \hat{\epsilon}(f_\theta, \tau'_j) = \frac{1}{m'_j}\sum_{k=1}^{m'_j}\mathbb{I}(g_{\phi'_j}(f_\theta(x_k)) \neq y_k),$$

where $\phi'_j$ is the parameters of $g_\phi$ learned from labeled samples $S'_j$ given $f_\theta$, and $m'_j = |S'_j|$ is the number of labeled samples in $S'_j$. $\mathbb{I}$ denotes the indicator function. Similarly, we can define the generalization error $\epsilon(f_\theta, \tau_i)$ and empirical error $\hat{\epsilon}(f_\theta, \tau_i)$ for training task $\tau_i$. With these definitions, we will proceed by introducing an auxiliary lemma, and then stating our main theoretical results. All detailed proofs of the following lemma and theorems are in the Appendix.

**Lemma 1.** *Let $\mathcal{A}$ be the attribute space, $L$ be the number of attributes. Assume all attributes are independent of each other given the class label, i.e. $p(a|y) = \prod_{l=1}^{L} p(a^l|y)$. For all $a_i \in \mathcal{A}$ and any two categories $y_k, y_t$, the following inequality holds:*

$$\sum_{a_i \in \mathcal{A}} |p(a_i|y_k) - p(a_i|y_t)| \leq d(y_k, y_t) + \Delta, \tag{4}$$

*where $d(y_k, y_t)$ is the distance as defined in Eq. (1) and $\Delta = \sum_{a_i \in \mathcal{A}} \frac{1}{2L}\sum_{l=1}^{L}(p(a_i^l|y_k) + p(a_i^l|y_t))$.*

Lemma 1 says that under the attribute conditional independence assumption, the distance between two attribute conditional distributions $p(a|y_k), p(a|y_t)$ over attribute space $\mathcal{A}$ is bounded by the defined distance $d(y_k, y_t)$ and a non-negative term $\Delta$. This result enables us to make use of the defined TAD metric in deriving the generalization error bound, leading to the following theoretical results.

**Theorem 1.** *With the same notation and assumptions as in Lemma 1, let $\mathcal{H}$ be the hypothesis space with VC-dimension $d$, $f_\theta$ and $g_\phi$ be the meta-learner and the base-learner as introduced in Sec. 4.2 respectively. Denote $g_{\phi^*}$ as the best base-learner on some specific tasks given a fixed meta-learner $f_\theta$. For any single training task $\tau_i = (\mathcal{D}_i, S_i)$ and a novel task $\tau'_j = (\mathcal{D}'_j, S'_j)$, suppose the number of categories in the two tasks is the same, then with probability at least $1 - \delta$, $\forall g_\phi \circ f_\theta \in \mathcal{H}$, we have*

$$\epsilon(f_\theta, \tau'_j) \leq \hat{\epsilon}(f_\theta, \tau_i) + \sqrt{\frac{4}{m_i}(d\log\frac{2em_i}{d} + \log\frac{4}{\delta})} + d_\theta(\tau_i, \tau'_j) + \Delta' + \lambda, \tag{5}$$

*where $\lambda = \lambda_i + \lambda'_j$ is the generalization error of $g_{\phi^*}$ and $f_\theta$ on the two tasks, i.e., $\lambda_i = \mathbb{E}_{(x,y)\sim\mathcal{D}_i}[\mathbb{I}(g_{\phi_i^*}(f_\theta(x)) \neq y)]$, $\lambda'_j = \mathbb{E}_{(x,y)\sim\mathcal{D}'_j}[\mathbb{I}(g_{\phi'_j^*}(f_\theta(x)) \neq y)]$. $\Delta'$ is a term depending on learned base-learners $g_{\phi_i}, g_{\phi'_j}$ and the best base-learners $g_{\phi_i^*}, g_{\phi'_j^*}$.*

Theoretically, the generalization error on a novel task is bounded by the the training task empirical error $\hat{\epsilon}(f_\theta, \tau_i)$ plus four terms: the second term $d_\theta(\tau_i, \tau'_j)$ is the model-related distance between $\tau_i$ and $\tau'_j$, which is derived based on Lemma 1; the third term $\Delta'$ reflects the classification ability of $g_\phi$, which converges to zero if the learned base-learners are equal to the best ones for both tasks; the last term $\lambda$ is the generalization error of $f_\theta$ and $g_{\phi^*}$, which depends on the attribute discrimination ability of $f_\theta$ and the hypothesis space of base-learner $g_\phi$. For a reasonable hypothesis space, if $f_\theta$ has a good attribute discrimination ability on both tasks, the last term usually converges to zero. Next we generalize this bound to the setting of $n$ training tasks.

**Corollary 1.** *With the same notation and assumptions as Theorem 1, for $n$ training tasks $\{\tau_i\}_{i=1}^n$ and a novel task $\tau'_j$, define $\hat{\epsilon}(f_\theta, \tau_{i=1}^n) = \frac{1}{n}\sum_{i=1}^n \hat{\epsilon}(f_\theta, \tau_i)$, then with probability at least $1 - \delta$, $\forall g_\phi \circ f_\theta \in \mathcal{H}$, we have*

$$\epsilon(f_\theta, \tau'_j) \leq \hat{\epsilon}(f_\theta, \tau_{i=1}^n) + \frac{1}{n}\sum_{i=1}^n\sqrt{\frac{4}{m_i}(d\log\frac{2em_i}{d} + \log\frac{4}{\delta})} + \frac{1}{n}\sum_{i=1}^n d_\theta(\tau_i, \tau'_j) + \Delta' + \lambda, \tag{6}$$

*where $\lambda = \frac{1}{n}\sum_{i=1}^n \lambda_i + \lambda'_j$, and $\Delta'$ is a term depending on the learned base-learners $\{g_{\phi_i}\}_{i=1}^n, g_{\phi'_j}$ and the best base-learners $\{g_{\phi_i^*}\}_{i=1}^n, g_{\phi'_j^*}$.*

Corollary 1 is a straightforward extension of Theorem 1, in which we consider multiple training tasks instead of a single training task. In Corollary 1, the generalization error on a novel task $\tau'_j$ is bounded partially by the average distance between task $\tau'_j$ and $n$ training tasks. Note that the distance $d_\theta(\tau_i, \tau'_j)$ used in the bound is model-related. Next, we further derive the relationship between model-related distance and model-agnostic distance as follows.

**Theorem 2.** *With the same notation and assumptions as in Corollary 1, assume that the conditional distribution $p(x|a^l)$ is task agnostic. If the number of labeled samples $m_i$ in $n$ training tasks and the number of labeled samples $m'_j$ in novel task $\tau'_j$ tend to be infinite, the following equality holds:*

$$\frac{1}{n}\sum_{i=1}^{n} d_\theta(\tau_i, \tau'_j) \leq \frac{1}{n}\sum_{i=1}^{n} d(\tau_i, \tau'_j). \tag{7}$$

Theorem 2 shows that when the number of labeled samples in $n$ training task and a novel task tends to be infinite, the model-related average distance can be bounded by the average TAD. Based on the universal approximation theorem [21, 15] and the same assumption as in Theorem 2, if $g_\phi \circ f_\theta$ is a complex enough network, the second and fourth terms on the r.h.s. of Eq. (6) in Corollary 1 will converge to 0, the first term $\hat{\epsilon}(f_\theta, \tau_{i=1}^n)$ and $\frac{1}{n}\sum_{i=1}^{n}\lambda_i$ will converge to Bayes risk [13]. We assume the Bayes risk is small enough, so that it can be ignored. In this case, the generalization error on novel task $\tau'_j$ is only bounded by $\lambda'_j$ and the average TAD, *i.e.*, $\epsilon(f_\theta, \tau'_j) \leq \lambda'_j + \frac{1}{n}\sum_{i=1}^{n} d(\tau_i, \tau'_j)$. Note that the average TAD is independent of models, thus we can rely solely on the TAD metric to measure the adaptation difficulty of each novel task, without the need for training any models.

## 5 Computing the Model-Agnostic Task Distance

### 5.1 Distance Approximation

In this section, we discuss practical considerations in computing the TAD on real data, where the attributes often take on discrete and finite values. For continuous and infinite attributes, it is possible to divide their values into many segments or discrete parts. Therefore, we denote $V^l$ as a finite set of possible values of attribute $a^l$, and re-express the TAD in Eq. (2) as

$$d(\tau_i, \tau'_j) = \frac{1}{2L|M|} \sum_{e_{kt} \in M} \sum_{l=1}^{L} \sum_{v \in V^l} \left| P^i_{a^l|y_k}[v] - P^j_{a^l|y_t}[v] \right|. \tag{8}$$

Computing the above distance is challenging as it requires finding the maximum matching $M$ with minimum weights, which is a combinatorial optimization problem. The Hungarian algorithm [25] is commonly used to solve the matching problem, but it is computationally expensive. Due to the high computational cost, we do not calculate this distance directly but estimate the approximation instead:

$$d(\tau_i, \tau'_j) \approx \frac{1}{2LC} \sum_{l=1}^{L} \sum_{v \in V^l} \left| \sum_{k=1}^{C} P^i_{a^l|y_k}[v] - \sum_{t=1}^{C} P^j_{a^l|y_t}[v] \right|, \tag{9}$$

where we assume all tasks have $C$ categories, which holds in most FSL settings. In Eq. (9), we simplify the comparison of attribute conditional distributions between two tasks by replacing the individual differences with the average difference. Through the above approximation, we can avoid seeking the minimum weight perfect matching $M$ and estimate the TAD efficiently.

### 5.2 Attributes Auto-Annotation

As stated before, our proposed theory and TAD metric rely on category-level attribute annotations that are possibly unavailable for some few-shot benchmarks, like *mini*ImageNet [48]. To address this issue, we propose a solution to auto-annotate attributes using a pretrained CLIP [36] model, which has demonstrated impressive zero-shot classification ability. We first pre-define 14 attribute labels based on pattern, shape, and texture. Then we create two descriptions for each attribute, such as "a photo has shape of round" and "a photo has no shape of round", which are used as text inputs for CLIP. We formulate the annotation problem as 14 attribute binary classification problems for each image using CLIP model. After that, we gather the attribute predictions of all images within the same category, which provides rough category-level attribute information while greatly reduces the cost of attribute annotations. With the above auto-annotation process, we can obtain the category-level attributes for *mini*ImageNet in just 4 minutes.

# 6 Experiments

In this section, we conduct experiments to validate our theoretical results and explore the generality of the proposed TAD metric in measuring task adaptation difficulty. Firstly, we evaluate our TAD metric in a scenario where human-annotated attributes are available and the training/novel tasks are sampled from the same dataset but with different categories. Next, we test the TAD in a more general and challenging scenario where human-annotated attributes are not available and the training tasks are constructed by sampling categories from different datasets.

## 6.1 Setups

**Datasets.** We choose three widely used benchmarks: (1) CUB-200-2011 (**CUB**) [49]: CUB is a fine-grained dataset of birds, which has 200 bird classes and 11,788 images in total. We follow [23] to split the dataset into 100 training classes, 50 validation classes and 50 test classes. As a fine-grained dataset, CUB provides part-based annotations, such as beak, wing and tail of a bird. Each part is annotated by a bounding box and some attribute labels. Because the provided attribute labels are noisy, we denoise them by majority voting, as in [24]. After the pre-processing, we acquire 109 binary category-level attribute labels. (2) SUN with Attribute (**SUN**) [34]: SUN is a scene classification dataset, which contains 14,340 images for 717 scene classes with 102 scene attributes. In following experiments, We split the dataset into 430/215/72 classes for training/validation/test, respectively. (3) *mini***ImageNet** [48]: *mini*ImageNet is a subset of ImageNet consisting of 60,000 images uniformly distributed over 100 object classes. Note that, different with CUB and SUN, *mini*ImageNet does not provide the attribute annotations, which means category-level attribute annotations are not available. Following [10], we consider the *cross-dataset scenario* from *mini*ImageNet to CUB, where we use 100 classes of *mini*ImageNet as training classes, and the 50 validation and 50 test classes from CUB.

**Attribute Prototypical Network.** Our theoretical analysis is based on a specific meta-learning framework with attribute learning (proposed in Sec. 4.2). Thus, we first instantiate a simple model under that framework as an example to verify our theory. Specifically, we adopt a four-layer convolution network (Conv-4) with an additional MLP as the meta-learner $f_\theta$. The convolutional network extracts feature representations from images, then the MLP takes features as input and predicts attribute labels. For base-learner $g_{\phi_i}$ parameterized by $\phi_i$, we simply choose an non-parametric base-learner like ProtoNet [42], which takes the attributes outputted by $f_\theta$ as input to calculate cosine distance between test samples and attribute prototypes, then predicts the target label. We call this method as *Attribute Prototypical Network* (APNet). We train APNet by simultaneously minimizing the attribute classification loss and the few-shot classification loss.

**Other FSL Methods.** Besides of APNet, we choose five classical FSL methods in the following experiments, since they cover a diverse set of approaches to few-shot learning: (1) Matching Network (MatchingNet) [48], (2) Prototypical Network (ProtoNet) [42], (3) Relation Network (RelationNet) [45], (4) Model Agnostic Meta-Learning (MAML) [17] and (5) Baseline++ [10]. Note that these FSL methods, unlike the above APNet, do not use attribute annotations during training. See more implementation and experimental details in the Appendix.

## 6.2 Task Attribute Distance with Human-annotated Attributes

**Approximate results with limited samples.** According to Theorem 2, if the number of labeled samples in tasks tends to be infinite, the TAD serves as a metric to measure the adaptation difficulty on a novel task. However, it is impossible that each task contains infinite labeled samples in real-world applications. Thus, we try to empirically verify the approximate theoretical result with limited labeled samples for each task. We train our APNet on CUB and SUN dataset, then use the provided attribute annotations to calculate the average distance $\frac{1}{n}\sum_{i=1}^{n} d(\tau_i, \tau_j')$ between each novel task $\tau_j'$ and $n$ training tasks $\{\tau_i\}_{i=1}^n$, respectively. For simplicity, we sample $n = 100,000$ training tasks to estimate the distances to $2,400$ novel tasks. Following $N$-way $K$-shot setting, each task only contains $N * K$ labeled samples for training or fine-tuning models. Fig. 4 shows the task distance and the corresponding accuracy on novel tasks for APNet. We can observe in Fig. 4 that as the distance increases, the accuracy of APNet decreases in both 5-way 1-shot and 5-way 5-shot settings on CUB and SUN. These results verify that TAD can characterize model's generalization error on each novel task and measure the task adaptation difficulty effectively, even with limited labeled samples for

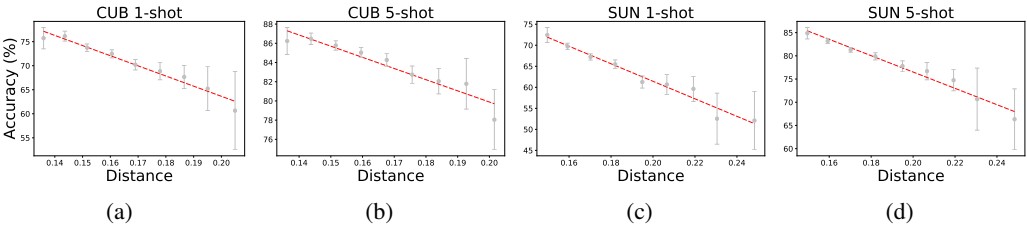

(a)  (b)  (c)  (d)

Figure 2: Accuracy of APNet in terms of the average task distance. (a)-(b) 5-way 1-shot and 5-shot on CUB. (c)-(d) 5-way 1-shot and 5-shot on SUN. In (a)-(d), each gray point denotes the average accuracy for all points in a distance interval, and the error bar denotes the confidence interval at 95% confidence level. The red dashed line is a fitted line, which shows the tendency of gray points.

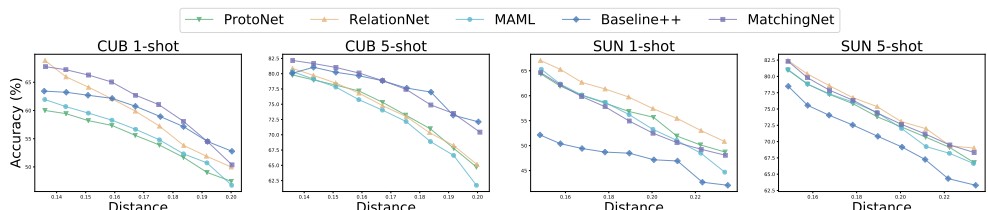

Figure 3: Accuracy of different methods in terms of the average task distance. From left to right, 5-way 1-shot and 5-shot on CUB/SUN. Each point denotes the average accuracy in a distance interval.

training and fine-tuning models. Note that the confidence interval in Fig. 4 is much larger for the last a few points. We argue this is because these distance intervals contain fewer novel tasks, thus the average accuracy of novel tasks in the interval is more easily affected by random factors.

**The number of labeled samples.**    Additionally, in Fig. 4, when comparing 1-shot and 5-shot results with the same distance interval, we find that the increase of accuracy varies at different distance intervals. For instance, on the CUB dataset (comparing Fig. 2(a) with Fig. 2(b)), when the task distance is 0.14, APNet shows an improvement of approximately 10% in accuracy for the 5-shot setting over the 1-shot setting, whereas it shows an improvement of around 15% for the distance of 0.20. This suggests that increasing the number of labeled samples is more effective for harder tasks. One possible explanation is that as the task distance increases, less knowledge can be transferred from training tasks, making it harder for models to adapt to the novel task. Hence, more information from the novel task is required for adaptation, and the model's performance can get more benefit from additional labeled samples.

**Other FSL models.**    We have shown that the average TAD can effectively reflect the task adaption difficulty for APNet. A natural question is whether the calculated task distance can be directly applied to other FSL methods, even if they do not follow the specific meta-learning framework and do not use the attributes during training. We try to empirically explore this question. With the same distance estimation and experimental setting, we conduct experiments with five different FSL methods on CUB and SUN. Fig. 3 shows the task distance and the corresponding accuracy of 2,400 novel test tasks for them. We observe similar results that with the increase of task distance, the accuracy of all FSL models tends to decrease. This indicates that even though these FSL methods do not use the attribute annotations during training, they implicitly learn mixture of attributes in their representations. Therefore, attribute-based distance can still reflect the task adaptation difficulty for them. These results demonstrate the generality of the proposed TAD metric, and provide some insight into the transferable knowledge that different FSL models have learned.

### 6.3    Task Attribute Distance with Auto-annotated Attributes

**Cross-dataset generalization.**    Another interesting question is how to utilize the proposed TAD metric in situations where human-annotated attributes are either unavailable or expensive to obtain. One classic example is the cross-domain few-shot learning problem, since it requires the annotation

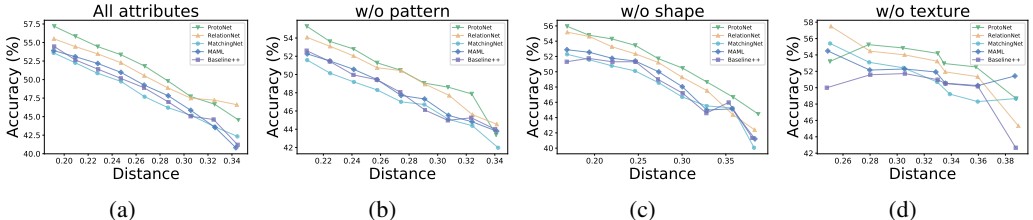

|        | All attributes | w/o pattern | w/o shape | w/o texture |
|--------|----------------|-------------|-----------|-------------|
| (a) | (b) | (c) | (d) |

Figure 4: Accuracy of different methods in terms of the average task distance with (a) all attributes, (b) removal of pattern attributes, (c) removal of shape attributes, (d) removal of texture attributes. The experiment is conduct in 5-way 5-shot setting.

of distinct datasets with a common attribute set. To address this challenge, we use the auto-annotation method described in previous section to annotate the attributes for both *mini*ImageNet and CUB. We then train the above FSL models on the cross-dataset scenario from *mini*ImageNet to CUB, and estimate the average task distance with the auto-annotated attributes. Fig. 4(a) illustrates the distance and corresponding accuracy of 2,400 novel test tasks with the auto-annotated attributes. We can find that, with the increase of task distance, the accuracy of different models tends to decrease. This phenomenon is consistent with our previous findings and shows the proposed TAD metric still works with auto-annotated attributes. Besides, we selectively remove some attributes to explore the influence of them in the distance-accuracy curve. Figs. 4(b) to 4(d) show that the exclusion of pattern, shape, and texture attributes exhibits varying degrees of influence on the decreasing tendency. Notably, we discover that texture attributes are more importance than others, as indicated by the more pronounced fluctuations in the curve.

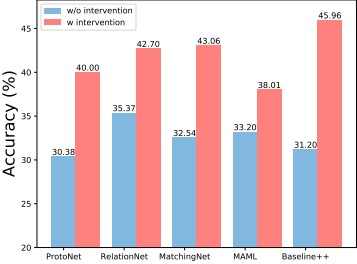

Figure 5: 1-shot results of test-time intervention (*mini*ImageNet→CUB). We only show the accuracy of the intervened tasks.

|              | 100 classes | 64 classes | $\Delta$ |
|--------------|-------------|------------|----------|
| **MatchingNet** | 50.0    | **51.7**   | -1.7     |
| **ProtoNet**    | **53.7**| 52.6       | 1.1      |
| **RelationNet** | **52.6**| 50.3       | 2.3      |
| **MAML**        | **51.6**| 49.7       | 1.9      |
| **Baseline++**  | 52.4    | **55.5**   | -3.1     |

Table 1: Comparison of 5-shot accuracy for different methods, when training on selected 64 classes instead of all 100 classes. $\Delta$ represents the difference in accuracy between the two cases. The best accuracy of each method is marked in bold.

**Test-time Intervention.** With the proposed TAD metric, we can quantify the adaptation difficulty of novel test tasks without training and adapting a model. This makes sense in real-world scenarios, as we can identify more challenging tasks before training and take interventions to improve a model's performance on those tasks. In this part, we explore a simple test-time intervention operation that supplies more labeled samples for harder novel tasks. To construct the intervention experiment, we first calculate the distances between each novel task and $n$ training tasks, as done in previous experiments. We manually set a constant threshold value, denoted as $r$, to identify the harder tasks that exhibit large distances and generally yield low accuracy. Once we identify these challenging tasks, we intervene by providing additional labeled samples, which contain more task-specific information. Specifically, in the 5-way 1-shot setting, we offer 4 extra labeled samples for each intervened task, which can be seen as the additional annotated samples. We set the distance threshold $r = 0.29$ for the cross-dataset scenario. With the above threshold values, we only need to intervene with a small subset of novel tasks (approximately 5%-7%), which greatly reduces the cost of sample annotation. The results of the test-time intervention can be seen in Fig. 5. As illustrated in Fig. 5, the test-time intervention can significantly improve the performance of different FSL models on the challenging novel tasks. Note that we are not proposing a novel intervention method, but demonstrating a practical application of the proposed TAD metric.

**Training with partial versus all tasks.** Using the proposed TAD, we can also investigate whether training on all available training tasks leads to better generalization compared to training on only the most related ones. While it is generally expected that training with more tasks helps to improve generalization, it remains a question whether this holds for heterogeneous data (e.g. cross-dataset scenario). To explore this, we follow the cross-dataset experiment setup from *mini*ImageNet to CUB, and select these less related training tasks based on their calculated task distances. We collect the classes in the selected tasks, compute the frequency of each class, and remove the most frequent 36 classes (out of a total of 100 classes in *mini*ImageNet). We then reconstruct training tasks based on the remaining 64 classes in *mini*ImageNet and retrain five FSL models. The results are presented in Tab. 1. It can be observed that incorporating more heterogeneous data that is less related to the novel tasks may not lead to much improvement ( for ProtoNet, RelationNet and MAML), but could even result in performance degradation ( for MatchingNet and Baseline++). This result may inspire future research on how to better utilize heterogeneous data to improve the performance of models in the cross-domain few-shot learning problem.

## 7 Conclusion

We propose a novel distance metric, called Task Attribute Distance (TAD), in FSL to quantify the relationship between training and novel tasks, which only relies on the category-level attribute annotations. We present a theoretical analysis of the generalization error bound on a novel task with TAD, which connects task relatedness and adaptation difficulty theoretically. Our experiments demonstrate TAD can effectively reflect the task adaptation difficulty for various FSL methods, even if some of them do not learn attributes explicitly or human-annotated attributes are not available. We also present two potential applications of our TAD metric: intervening a small set of difficult novel tasks before training, and investigating the benefits of heterogeneous training tasks. We believe our theoretical and empirical analysis can provide more insight into few-shot learning and related areas.

**Limitations and broader impact.** Our theory and proposed TAD metric assume that attributes are conditionally independent without considering the correlations that may exists between them. Furthermore, identifying which attributes are critical in the distance metric is still an open question. However, we believe that our work lays a solid foundation for measuring task relatedness and adaptation difficulty of novel tasks, which offers a starting point for further research in few-shot learning and related areas.

## Acknowledgments

This work is partially supported by National Key R&D Program of China no. 2021ZD0111901, and National Natural Science Foundation of China (NSFC): 61976203 and 62276246.

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
