# OpenReview forum: "Understanding Few-Shot Learning: Measuring Task Relatedness and Adaptation Difficulty via Attributes"
_NeurIPS.cc/2023/Conference — NeurIPS 2023 poster_

### Official Review · Reviewer_b4dX · 2023-06-22

**Soundness:** 3 good
**Presentation:** 4 excellent
**Contribution:** 2 fair
**Rating:** 6
**Confidence:** 5

**Summary:**

The paper proposes a new metric, the Task Attribute Distance (TAD), to quantify the link between two tasks for FSL. The distance is based on the relationship between some pre-defined attributes and classes in the two tasks. This distance can be computed from a model point of view, considering the distribution of attributes in the features space of a given model, or in a model-agnostic way. The paper presents two main theoretical results about TAD:
- first, showing the link between TAD from a model computed on training tasks and a given novel task, and the generalization error on this novel task.
- second, the link between the model-related TAD and the model-agnostic one.
Then the paper discusses how to empirically estimate the model-agnostic TAD, and how to create attributes automatically for datasets without ones, using a pretrained CLIP model.
Finally, the paper provides experiments on two datasets with attributes labels (SUN and CUB), and one with attributes from auto-annotation (mini-imagenet). The results show that the task distance is correlated with the performance of the models. A final experiment shows that some models benefits from removing frequent classes in training.

**Strengths:**

- *Originality:* the metric presented is novel and represents an interesting way to measure task difficulty. It uses information about attributes related between the tasks, as opposed to previous works which mainly used model's outputs.
- *Quality:* The claims are all supported, and the paper presents theoretical results with proofs to link the TAD metric with generalization bounds. The experimental results show the correlation between the distance and the performance. I took a look at the proofs and the results are sounded.
- *Clarity:* The paper is clearly written and easy to follow.

**Weaknesses:**

The main weakness of the paper is the significance and the quality of the analysis regarding the attributes automatically obtained.
- I found the metric interesting because it can be computed in a model-agnostic way, but its main problem is that it relies on attributes that are not always available on datasets. Even though the paper explains a way to automatically annotate with attributes, an evaluation of the quality of the obtained attributes is missing. We can even see on Figure 4 that the correlation between the obtained attributes and the performance is less clear than on other datasets. It would also be interesting to show more examples of the obtained attributes, for a qualitative analysis. Because it relies too much on attributes, and on their quality, I am afraid this metric will be of limited impact.
- I think the paper is missing an application of the proposed metric in a setting where it could be useful. Besides the experiments showing the link between the distance and the performance, I'm not really convinced by the last experiment on training with less training tasks. I don't think it is a good representative of potential applications.

**Questions:**

- I would like to see preliminary results for potential applications of this metric to assess the significance of the work.
- What are the attributes obtained for mini-Imagenet ? Are they sounded ?

**Limitations:**

The authors have included a discussion about the limitations of their works. I don't think that a discussion on societal impact is needed for this work.

---

> ### Author Rebuttal · Authors · 2023-08-09
>
> We thank you for taking the time out to review our work. We are glad that you found the proposed metric to be both novel and interesting, and the represented proofs to be sounded.
> In the following we address your concerns and questions in the order they are asked.
>
> **W1: An evaluation of the quality of the obtained attributes is missing. It would also be interesting to show more examples of the obtained attributes for a qualitative analysis.**
> Thanks for pointing out this question. We try to evaluate the quality of the auto-annotated attributes and then give some examples for qualitative analysis.
> + **Auto-annotation evaluation.** Due to the absence of attribute annotations in the miniImagenet dataset, we collect the annotations ourselves. We first pre-define 25 attribute labels, then random select 50 images and proceed to annotate them. By comparing the ground-truth annotations and the results produced by CLIP model, we discern that the average accuracy across the 25 attributes approaches 0.65. Notably, we observe that CLIP achieves good performance across the majority of attributes, with accuracy from 0.7 to 0.9. However, it fails on some attributes, such as "white," "pink","smooth," and "shiny", where the accuracy decreases to approximately 0.2. This discrepancy could potentially stem from the less frequency of these attributes in the CLIP training dataset. To understand the underlying factors more comprehensively, further experiment is required.
> + **Qualitative examples.** Fig.3 (in the pdf file of global response) shows the qualitative examples of the obtained attributes. We find that CLIP model can annotate the main attributes correctly in an image. The common wrong cases are color attributes. CLIP model always predicts more colors than manual annotations.
>
> **W2&Q1: I think the paper is missing an application of the proposed metric. I would like to see preliminary results for potential applications of this metric to assess the significance of the work.**
> We present two potential applications in the last experiment and Appendix 4.2.
> Specifically, in Appendix 4.2, we design a toy intervention experiment.
> This intervention experiment is motivated by the property of TAD metric that we can quantify the adaptation difficulty of novel test tasks without the need to train a model.
> This property enables us to identify more challenging tasks before training and take interventions to improve a model's performance on those tasks.
> In the intervention experiment, we intervene these challenging tasks by simply providing additional labeled samples.
> With a proper distance threshold, we only need to intervene with a small subset of novel tasks (approximately 5%-7%), which greatly reduces the cost of sample annotation and improves the performance of various models.
> Furthermore, inspired by the question raised by Reviewer mMtZ, we conceive another possible application for our proposed metric: it can be used to explore which tasks are not practical to do few-shot classification at all.
> More details about this application can be found in our response to Weakness 2 for Reviewer mMtZ.
>
> **Q2: What are the attributes obtained for mini-Imagenet? Are they sounded?**
> In the attribute auto-annotation process, we follow previous work [ref1] and pre-define 25 attribute labels. These 25 attributes are semantic appearance attributes, including color, pattern, shape and texture. The details about these attributes are described in the following table.
>
> | | attributes |
> |----------|------------|
> | Color    | black, blue, brown, gray, green, orange, pink, red, violet, white, yellow           |
> | Pattern  | spotted, striped           |
> | Shape    | long, round, rectangular, square           |
> | Texture  | furry, smooth, rough, shiny, metallic, vegetation, wooden, wet           |
>
> [ref1] Attribute learning in large-scale datasets. In ECCV 2010.

---

> > ### Comment · Reviewer_b4dX · 2023-08-14
> >
> > Thank you for the detailed answers and rebuttal.
> > - The detailed analysis provided, both qualitative and quantitative, of the auto-annotation process has cleared my concerns. I appreciate the comparison with manual annotation. The majority of predicted attributes are sounded, even though there seem to be some confusion with the background.
> > - I find the application of finding difficult tasks or difficult domains for annotation more significant than the "partial training" presented in 6.3. Even though the experiments in Appendix 4.2 could be more convincing with a comparison of intervention using different metrics, similarly to the table presented in the rebuttal, I think this kind of application can be useful in active learning scenarios.
> >
> > All my concerns have been addressed by the authors in their answers and rebuttal. I'm raising my score to "weak accept". I highly encourage the authors to add these discussions in the revised version.

---

### Official Review · Reviewer_mMtZ · 2023-06-30

**Soundness:** 3 good
**Presentation:** 3 good
**Contribution:** 3 good
**Rating:** 7
**Confidence:** 3

**Summary:**

This work raises two questions regarding the setting of few-shot classifications: (1) How to evaluate the difference between training and novel tasks? (2) How does the difference impact FSL methods? For the first question, the authors proposed Task Attribute Distance (TAD), which uses attributes from CLIP to evaluate task differences. For the second question, the author derives upper bounds for the task and show some empirical evaluation.

**Strengths:**

The author provides some theoretical insight into the difficulty of few-shot classifications with clear writing.

**Weaknesses:**

1. It is unclear how the manually selected 25 attribute impact the performance. The author also admits that the 25 pre-defined attributes are not comprehensive enough for the experiment in Sec 6.3.
2. It is not very intuitive how this work can provide insight into few-shot learning areas. For example, for the experiments conducted in current few-shot papers, what is their average TAD? Can we tell the difficulty of these tasks based on their TAD? If some settings have TAD larger than a threshold, can we tell that these settings are not practical to do few-shot classification at all? It will be much more impactful if the author can point out problems or expectations of the few-shot learning area based on their theorem.
3. Minor errors: Line 291, ProtoNet calculated Euclidean distance. It is MatchingNet that calculates cosine distance.


**Questions:**

1. It is interesting that Figure 2 shows an almost linear response between distance and accuracy. Is this expected from the theory?
2. Comparing Figures 2 and 3, it seems that the accuracy of APNet is generally better than other few-shot methods. Does that mean the APNet, which uses additional information from CLIP models, can be considered as an upper-bound method so we know which tasks have reached their best performance and we should not further push it? Can the authors evaluate it?



**Limitations:**

The author addressed the limitation in the selection of attributes, as I mentioned in the first point in the weakness section. Also, while the author raises several interesting observations and tries to derive the theory behind it, there is still some gap between the theory and the experiments, as I mentioned in the questions sections.

---

> ### Author Rebuttal · Authors · 2023-08-09
>
> Thank you for taking the time to review our work.
> We are glad that you were interested in our observations, and we appreciate the suggestions you raised to enhance the impact of our paper.
> In the following we address your concerns and questions in the order they are asked.
>
> **W1: It is unclear how the manually selected 25 attribute impact the performance.**
> We first explain how we select the 25 attributes in the auto-annotation process, then we perform a sensitivity analysis to explore the impact of these attributes.
> + **Attribute selection and universality.** In the process of attribute selection, we follow previous work [ref1] and pre-define 25 semantic appearance attributes. A comprehensive list of these attributes can be found in the response to Question 2 for Reviewer b4dX. These 25 attributes encompass a wide spectrum including color, pattern, shape and texture, thus can be applicable across various datasets.  Fig. 4 showcases the effectiveness and universality of these attributes on miniImagenet and CUB datasets, although the decreasing tendency is not a strict monotonic pattern.
> + **Sensitivity analysis.** To explore the influence of the selected 25 attributes, we exclude some attributes and observe the changes in distance-accuracy curve.
> Fig.2 (in the pdf file of global response) shows that the exclusion of 11 color attributes makes the curve more linear, consistent with the linear tendency of Fig.2 and Fig.3 in main paper. A possible explanation could be that the auto-annotated color attributes are more susceptible to the background.
> Furthermore, the exclusion of pattern, shape, and texture attributes exhibits varying degrees of influence on the decreasing tendency.
>
> [ref1] Attribute learning in large-scale datasets. In ECCV 2010.
>
> **W2: How this work can provide insight into few-shot learning areas? It will be much more impactful if the author can point out problems or expectations of the few-shot learning area based on their theorem.**
> Thanks for this valuable suggestion, and we will point out some expectations of few-shot learning area based on the insights gleaned from our theorem:
> + **Explore the boundaries of few-shot learning.** Based on our theorem and proposed metric, we can explore which tasks are not practical to do few-shot learning at all. Taking CUB dataset as an example. In Fig.3 (left, 5-way 1-shot setting), assuming a consistent tendency for the accuracy-distance curve, the accuracy of various FSL models reduces to 0.2 when the distance increases from 0.3 to 0.4. In other words, different FSL models performs like random guess when the distance of a novel task exceeds the threshold 0.4. In such case, these tasks are not practical to do few-shot classification at all, which is beyond the boundary of few-shot learning problem.
> We conducted empirical verification of this idea by first selecting tasks with the highest distances and then evaluating the performance of FSL models on these tasks. Notably, as the distance of novel tasks approaches 0.3, the accuracy of different FSL models reduces to the range of 0.25-0.35, aligning with the observed curve in Fig.3.
> + **Explore the utilization of heterogeneous data in cross-domain few-shot learning**. Based on our theorem and metric, a viable direction in cross-domain few-shot learning is to select appropriate training tasks to optimize models, like [25]. As shown in Sec 6.3, our study reveals that incorporating more training tasks that are less related to novel tasks, may not lead to much improvement. It could even result in performance degradation. This observation suggests that, with the context of cross-domain few-shot learning, selecting training tasks that are closely related to the novel tasks could be more beneficial for improving performance.
>
> **W3: Minor errors: Line 291, ProtoNet calculated Euclidean distance. It is MatchingNet that calculates cosine distance.**
> Thanks for pointing out this. It is not an error but a less clear description. At line 291, our intention is to describe a non-parametric base learner similar to ProtoNet, utilizing the average of sample embeddings to calculate the prototypes. We will revise this sentence to avoid misunderstandings in the final version.
>
> **Q1: It is interesting that Figure 2 shows an almost linear response between distance and accuracy. Is this expected from the theory?**
> Yes. At line 233, if $\lambda_j'$ is a small constant, we anticipate observing a linear correlation between the average TAD and the generalization error bound.
> It is noted that Theorem 2 and the following analysis rely on the assumption of infinite labeled sample, whereas we empirically verify the linear relationship using limited samples for each task.
> This emphasizes that the infinite sample assumption can be further relaxed to closely align with the experiment finding.
>
> **Q2: Can the APNet be considered as an upper-bound method? So we know which tasks have reached their best performance and we should not further push it.**
> APNet utilizes additional attributes, whether human-annotated or auto-annotated, thus its accuracy is generally better than other few-shot methods.
> However, designating APNet as an upper-bound method requires more evaluation and analysis. This is because the best performance on each task depends on many factors, such as network architecture, training strategy and attribute annotation noise.
> For a comprehensive understanding of which tasks have reached their best performance and hence not necessitate to further push, an extensive evaluation and analysis considering all these factors (including APNet and many SOTA methods) are required, which lies beyond the scope of this paper.

---

> > ### Comment · Reviewer_mMtZ · 2023-08-17
> >
> > Thank the author for the detailed responses. I am now more convinced by the work and raise my score to "accept".

---

### Official Review · Reviewer_aTLp · 2023-07-06

**Soundness:** 3 good
**Presentation:** 2 fair
**Contribution:** 2 fair
**Rating:** 4
**Confidence:** 3

**Summary:**

The paper proposes a new metric called TAD (Task Attribute Distance) to measure the distance between two tasks which is expected to serve as a proxy for how difficult the target task is in comparison to the source pre-training data and also provide an approximation of the few-shot classification performance of a given task. The proposed method works by calculating a set of attributes for each class per dataset and then by computing an approximation of the maximum bipartite matching score between the classes from the source and target tasks. Authors provide theoretical justification related to the generalization bounds and also provide some empirical results.

**Strengths:**

* The proposed method has a unique viewpoint -- instead of measuring task distance by using gradients or the fisher information matrix like some other methods, using a set of attributes as the anchors and computing distance based on the learned values of those attributes is a novel idea.

* Given that not all datasets come with a set of pre-defined attributes, authors provide a practical backup solution to define a set of prompts per dataset where such attributes are not available and use CLIP to determine to get a score for each attribute -- after that, the original algorithm can be used.

* The paper provides theoretical guarantees regarding the generalization error on a new task in terms of the proposed metric.

* I appreciate the effort from authors to reduce the computational complexity of the bipartite matching by replacing the `hungarian` algorithm with a reasonable approximation.

**Weaknesses:**

* The practicality of this metric is questionable -- it needs a somewhat large set of attributes per dataset and for most of the datasets, such attributes are not available and it would require users to generate a set of prompts and then feed the data through a CLIP model and perform trial and error before finding the right set of attributes.

* Although in the related works section, the paper mentions a few works from the literature, I do not see any direct comparison between this metric and those in the experimental setup. It is hard to understand how good this metric is compared to other metrics.

* The paper here misses comparing against (both method wise and experiment wise) against a relevant paper called Task2Vec - https://arxiv.org/abs/1902.03545.

* The proposed metric is heavily tied to meta-learning algorithms and also where the number of ways (in K-way N-shot) have to be fixed between meta-train and test - which means to use this metric, one needs to train multiple meta learners with different number of ways. Please let me know if my understanding is incorrect.

**Questions:**

* How will this method work if there is partial overlap in terms of classes between train and test? Is it a strict condition that the classes have to be disjoint between train and test?

* How will the attribute prediction work for domains where CLIP may not work well?

* Why does the attributes have to be dataset specific and defined apriori? Can it be something that is learnable too?

For other questions, please see my comments under weaknesses section.

**Limitations:**

Authors mentioned some limitations of their work in the paper and those seem reasonable.

---

> ### Author Rebuttal · Authors · 2023-08-10
>
> Thank you for taking the time to review our work.
> We are glad that you deemed our method has a unique viewpoint, found TAD metric novel and recognized the effort we make to reduce the computational complexity.
> In the following, we address your questions and concerns in the order of being proposed.
>
> **W1: The practicality of this metric is questionable when human annotated attributes are unavailable.**
> We provide some strategies to improve the efficiency of auto-annotation thus mitigate the problem you raised:
> + **Domain-Specific Prompts.** Existing works [ref] offer several domain-specific prompts to improve the zero-shot image classification ability of CLIP for different datasets. Incorporating these established domain-specific prompts considerably alleviates the burden of generating a set of new prompts.
> + **Coarse-Grained Attribute.** Pre-defining and auto-annotating a large set of fine-grained attributes incur substantial costs. A possible solution is to pre-define a small set of coarse-grained attributes, as adopted in our experiments, and it works to reflect the task adaptation difficulty.
>
> [ref] Learning to Prompt for Vision-Language Models. In IJCV 2022.
>
> **W2: Lack of comparisons between proposed TAD metric and other metrics by experiments.**
> Thanks for this valuable suggestion.
> For comparing different metrics, we design a task selection experiment.
> More specifically, we select top 5% novel tasks with the highest distances computed by different metrics, and
> then evaluate the accuracy of FSL models on these chosen tasks.
> The central hypothesis behind this experiment is that if a distance metric can better reflect task difficulty, then novel tasks with the highest distances should be more challenging.
>
> Due to the page limit, we present the experiment details and table in the global response.
> The table shows that TAD significantly outperforms other comparison methods in identifying more challenging novel tasks, demonstrating the effectiveness of TAD metric.
> Furthermore, we are glad to highlight that the computational efficiency of TAD surpasses other methods, as illustrated in the last column of table.
>
> **W3: The paper here misses comparing (both method wise and experiment wise) against a relevant paper called Task2Vec.**
> Thank you for pointing out this related work, we will include it in final version.
> In the following, we compare Task2Vec and our work from method wise and experiment wise.
> + **Method Wise.** Task2Vec encodes each task into an embedding, and uses the norm of embedding to reflect task complexity.
> Like ours, Task2Vec provides a way to measure the difficulty of a novel task. However, there are two main differences: (1)
> Task2Vec utilizes the second derivative of the loss on novel tasks, whereas TAD employs attribute distributions in both training and novel tasks. These two approaches can be combined together to obtain a more accurate measure.
> (2) Task2Vec requires retraining the classifier and computing fisher information matrix for each novel task, thus it is time-consuming.
> However, TAD measures the task difficulty efficiently without the need of training any model.
> + **Experiment Wise.**
> We conduct a task selection experiment for comparisons.
> As depicted in the table (see global response), Task2Vec works well with RelationNet and APNet, while fails with other FSL models.
> We attribute this to the reliance on a pretrained model.
> Moreover, Task2Vec suffers from the computational burden.
> It demands 100 minutes for 2400 novel tasks. In contrast, TAD completes the task difficulty measure in just 3 seconds.
>
> **W4: The proposed metric is not applicable with varying number of ways: one needs to train multiple meta learners.**
> We need to clarify that using TAD does ***not*** require training multiple meta learners.
> The TAD metric can be applied directly in the scenario by averaging the attribute conditional distributions of each task in Eq.9.
> To verify this point, we conduct experiments with varying numbers of ways.
> Fig.1 (in the pdf file of global response) confirms that TAD metric can handle this scenario.
>
> **Q1: How will this method work if there is partial class overlap between train and test?**
> The proposed TAD metric can be directly applied when a class overlap exists.
> From Eq.9, if a category in the training task $\tau_i$ also appears in the novel task $\tau_j'$, the corresponding attribute conditional distributions can be eliminated within the subtraction formula.
>
> **Q2: How will the attribute prediction work for domains where CLIP may not work well?**
> The attribute auto-annotation process may not work well on a dataset where the performance of CLIP is subpar.
> In light of this, we present solutions to tackle this problem:
> + **Parameter-Efficient Fine-Tuning.**
> An effective way is fine-tuning CLIP model to adapt general knowledge to specific datasets.
> To this end, we can augment CLIP with additional parameters (like prompts or adapters) and train only the newly introduced parameters with few labeled samples.
> + **In-Context Learning.** A better way is utilizing in-context learning capacity of pretrained vison-language models, such as GPT-4. In this approach, the labeled samples are treated as demonstrations. Take the demonstrations and query as input, GPT-4 can make predictions without fine-tuning models.
>
> **Q3: Why does the attributes have to be dataset specific and defined apriori?**
> We consider the dataset specific and predefined attributes due to two reasons:
> + **Interpretability.** Predefined attributes offer interpretability, enabling a better way to understand the transferable knowledge acquired by distinct FSL models.
> + **Model Independence.** Predefined attributes correspond to physical entities that are independent of models. Consequently, the distance metric derived from these attributes can be calculated without training a model.
>
> It is noted that learning these attributes in an unsupervised manner might compromise these advantages.

---

> > ### Comment · Reviewer_aTLp · 2023-08-14
> > **Thanks for your response**
> >
> > Thanks for your detailed response and also, including a new set of experiments by comparing different metrics. Most of my comments are addressed apart from two:
> > * What is the pre-trained model that you chose for `Task2Vec`? For fair comparison, it should be the same CLIP model (vision encoder of CLIP) that the proposed method here chooses and not ResNet-50. The comment that `Task2Vec` is reliant on a pre-trained model is also true for this method - it is reliant on a "pre-trained" CLIP model (which is trained on 400M+ samples as opposed to ResNet-50 trained on 1M ImageNet). We need to untangle the effect of the algorithmic contribution from the goodness of the pre-trained model.
> >
> > * The "Domain Specific Prompts" that was mentioned in the first response need to be curated per dataset and then has to go through trial and error to see how good it performs. I agree that it is somewhat of a mitigation but not a robust and complete one and hence my other suggestion to rather learn these attributes which authors seem to not prefer due to concerns around interpretability.

---

> > > ### Author Response · Authors · 2023-08-15
> > > **Thank you for your response!**
> > >
> > > Thank you for your response!
> > > We will address your questions and concerns in the following:
> > >
> > > **Q1: What is the pre-trained model that you chose for Task2Vec? It should be the same CLIP model for a fair comparison.**
> > >
> > > As indicated in the global response, we chose the ResNet-101 pretrained on ImageNet for Task2Vec.
> > > I appreciate your attention to this detail, and I would like to clarify that the comparison is still fair.
> > > We conducted the task selection experiment on CUB dataset, which provides 109 human-annotated attributes.
> > > Consequently, the calculation of TAD metric does not require the auto-annotated attributes, thus is not reliant on a pre-trained CLIP model.
> > >
> > > To further verify the effectiveness of TAD metric, we extend the task selection experiment into a cross-dataset scenario (from mini-Imagenet to CUB).
> > > In this scenario, human annotated attributes are unavailable, thereby a pre-trained CLIP model is used to annotate attributes for the calculation of TAD.
> > > Here are the results that compare Task2Vec with TAD using the same pre-trained CLIP model for them:
> > > ||MatchingNet|ProtoNet|RelationNet|Baseline++| Cost|
> > > |-|-|-|-|-|-|
> > > |Task2Vec|**-4.73**| -3.65|-4.79|**-5.83**|100min|
> > > |TAD|-2.31|**-5.95**|**-6.19**|-3.86| 3s|
> > >
> > > We can find that TAD metric can achieve comparable results with Task2Vec even when the human-annotated attributes are unavailable.
> > > Furthermore, the computational efficiency of TAD greatly surpasses Task2Vec, as illustrated in the last column of the table.
> > > TAD requires only 3 seconds to compute across 2400 novel tasks, while Task2Vec costs nearly 100 minutes.
> > >
> > > **Q2: The "Domain Specific Prompts" is a mitigation but not a robust and complete solution.**
> > >
> > > Thanks for the further question about the efficiency of the auto-annotation part. It's true that the process of selecting hand-crafted prompts for different datasets involves trial and error, thereby demanding much human effort. In light of this, we give a more robust and complete alternative to annotate attributes: introduce learnable vectors to represent context words (like prefix and suffix) while keeping the predefined attributes unchanged. This approach requires as few as one or two labeled samples to learn prompts [ref1, ref2] and does not need extra cost to select hand-crafted prompts with trial and error.
> > >
> > > [ref1] Learning to Prompt for Vision-Language Models. IJCV 2022.
> > > [ref2] Conditional prompt learning for vision-language models. CVPR 2022.
> > >
> > > **To conclude**, we thank you again for the effort you have invested in this review. If there is something you feel we have not adequately addressed yet, please do not hesitate to engage with us.

---

> > > > ### Author Response · Authors · 2023-08-19
> > > > **Look forward to your response**
> > > >
> > > > Dear Reviewer aTLp,
> > > >
> > > > We appreciate your time and effort in reviewing our work and offering valuable suggestions. We are looking forward to your response to see whether we have addressed your concerns. If you have any further questions regarding the paper, please do not hesitate to let us know.
> > > >
> > > > Thanks, Authors.

---

### Official Review · Reviewer_PSzT · 2023-07-07

**Soundness:** 4 excellent
**Presentation:** 4 excellent
**Contribution:** 3 good
**Rating:** 6
**Confidence:** 5

**Summary:**

This paper proposes a new metric--Task Attribute Distance (TAD)--to measure the distance between training and novel tasks in few-shot learning. TAD, based on pre-defined attributes of images, calculates a minimum matching between classes in training and novel tasks, where each edge in the bipartite graph is represented by the average TV distance between each two categories on all attribute variables. Using a meta-learning framework, the authors give a theoretical guarantee of the positive connection between TAD and adaptation difficulty. Experiments verify that an increased TAD can indeed lead to decreased few-shot learning performance, hence increased adaptation difficulty.

**Strengths:**

-  How to measure task similarity is central to few-shot learning problems. Using attributes to calculate task similarity is intuitive and makes sense to me, and has the advantage of not relying on specific models.
- It is clever to utilize CLIP to automatically annotate attributes.
- The theoretical analysis gives a guarantee of the effectiveness of the proposed metric.
- Experiments on in-domain and cross-domain few-shot learning problems verify that the proposed metric can indeed serve as a surrogate of task adaptation difficulty, independent of the model used.
- The writing is clear and easy to follow.

**Weaknesses:**

- The proposed metric only considers the distance between categories of two tasks, but not considers the relationship among categories inside each task. Thus some important characteristics of a task, like finegrainedness, are not taken into account.
- In Theorem 1, VC dimension can go to a large value for neural networks, so the second term is not negligible, and the whole bound is thus not tight.
- Theorem 2 and the following analysis depend on the assumption that the number of samples in the support set goes to infinity, thus not applied to few-shot learning, deviating from the main topic of the paper.
- The theorem can be applied only to the hand-crafted meta-learning algorithm.
- It seems to differ a lot switching from minimum matching to average difference.
- Automatic annotation may lead to wrong definitions and assignments of attributes. This problem manifests in Figure 4.



**Questions:**

- Why consider the TV distance?
- What if decreasing distance further in Figure 2-3, for example, below 0.1?

**Limitations:**

The authors have adequately stated the limitations.

---

> ### Author Rebuttal · Authors · 2023-08-09
>
> Thank you for the detailed and constructive comments.
> We are glad that you consider our studied problem is central in FSL problems and find the auto-annotation part clever.
> In the following, we address your questions and concerns in the order of being proposed.
>
> **W1: The proposed metric only considers the distance between categories of two tasks, but ignores the relationship among categories inside each task.**
> Yes, we focus on measuring the relationship between categories of two tasks while ignoring the internal relationship within each task.
> The rationale behind this idea is our belief that the former takes a more dominant role in the FSL problems.
> In the common 5-way setting, each novel task has only 5 categories, thereby the internal relationship is less influential.
> To account for the internal relationship, a possible solution is to measures the distance between attribute distributions of categories within a novel task, and then incorporate the distance into generalization error bound.
>
> **W2: The bound in Theorem 1 is not tight due to a large value of VC dimension for neural networks.**
> Thanks to point out this problem.
> We will modify the Eq.13 in the proof of theorem 1 (see Appendix) by replacing the VC dimension with Rademacher complexity, which is commonly used in generalization theory for a tighter bound.
> We would like to highlight that this modification will only impact the second term in Theorem 1, as well as Corollary 1.
> Note that our theoretical conclusions and experimental findings will remain unchanged after the modification.
>
> **W3: The assumption of infinite samples in support set for Theorem 2 and the following analysis deviates from the few-shot learning topic.**
> This is a nice question -- thank you for asking!
> Theorem 2 assumes infinite labeled samples to guarantee the proximity between the learned function $f_\theta$ and the true labeling function $f: \mathcal{X} \rightarrow \mathcal{A}$.
> To relax the assumption, we introduce the notation of $\epsilon$-approximation Network, as proposed in [ref1].
> The $\epsilon$-approximation Network is a model trained on support set such that the performance on query set is not less than 1-$\epsilon$, where $0<\epsilon<1$.
> Building upon this notation, we restate the Theorem 2 as:  With the same notation and assumptions as in Corollary 1, assume the conditional distribution $p(x|a^l)$ is task agnostic and $f_\theta$ is a $\epsilon$-approximation Network. The model-related average distance can be bounded by the average TAD, if $\epsilon$ tends to zero.
> In the final version, we will provide a more precise formulation of Theorem 2, and, if necessary, effectuate the modifications within the corresponding proof.
> [ref1] Task Affinity with Maximum Bipartite Matching in Few-Shot Learning. In ICLR 2022.
>
> **W4: The theorem can be applied only to the hand-crafted meta-learning algorithm.**
> Our theorem can be applied to other few-shot learning framework, such as transfer-based approach [9, ref2], with slight modification.
> More specifically, we can consolidate all few-shot training tasks as a single training task, containing all training categories and samples.
> Then replace the meta-learning framework discussed in Sec 4.2 with a transfer-based framework, which performs early pre-training on all training samples and subsequent fine-tuning on each novel task.
> For this new framework, Theorem 1 can be directly applied, which permits the applicability of our theorem into transfer-based few-shot approaches.
> Furthermore, how to incorporate other few-shot methods (like optimization-based approach) into our theoretical framework is an interesting problem, and we leave them into future work.
> [ref2] Meta-baseline: Exploring simple meta-learning for few-shot learning. In CVPR 2021.
>
> **W5: It seems to differ a lot switching from minimum matching to average difference.**
> We switch from minimum matching to average difference mainly due to the computational burden.
> To find a maximum matching, the commonly used Hungarian algorithm exhibits a time complexity of $O(V^2logV + VE)$ or $O(V^2E)$, where $V$ is the number of vertices and $E$ is the number of edges.
> Due to the high computational cost, we approximate it with average difference.
> This approximation preserves some properties in original TAD, such as symmetricity and identity, while significantly mitigating computational overhead.
> Empirical results demonstrate the reasonability of this approximation, as it can effectively reflect the task adaptation difficulty for various FSL methods.
>
> **W6: Automatic annotation may lead to wrong definitions and assignments of attributes.**
> To evaluate the quality of auto-annotated attributes, we manually annotate a few images ourselves.
> Detailed experiment results and analysis can be found in the response to Weakness 1 for Reviewer b4dX.
> To acquire more accurate annotations, a better way is to fine-tune CLIP model with a few human-annotated samples, which can be achieved by learning prompts or adapters.
>
> **Q1: Why consider the TV distance?**
> We consider the TV distance due to the following two advantages: (1) Bounded distance. TV distance has a bounded value that ranges from 0 to 1, thus is well-suited for establishing a tighter upper bound on generalization error. (2) Efficient computation. When the set of measurable subsets is finite, TV distance is equal to half of the $L_1$ distance. In real-world scenarios, attributes often take on discrete and finite values, enabling the efficient computation of the TV distance.
>
> **Q2: What if decreasing distance further in Figure 2-3, for example, below 0.1?**
> According to our theorem, we speculate that further decreasing the distance in Fig. 2-3 could lead to an increase in accuracy.
> However, empirically confirming this speculation poses challenges.
> In our experiments, we observe a minimal task distance of 0.134, and no tasks exhibit a distance smaller than 0.1 within the current datasets and settings.

---

> > ### Comment · Reviewer_PSzT · 2023-08-15
> >
> > Thanks for the authors' detailed response. I have looked through other reviewers' comments, as well as the authors' rebuttals. While I think the quality of this work has already met the acceptance bar, this work can be improved further in several ways (some have already been done during the discussions), which can make this paper even stronger. I decide to maintain the score.

---

### Author Rebuttal · Authors · 2023-08-10

We are grateful to all reviewers for the generous effort they have invested in reviewing our work, and are enthused to learn that they consider our studied problem is "central" in few-shot learning (Reviewer PSzT), find our ideas "unique", "novel" (Reviewer aTLp), "interesting" and "well supported" (Reviewer b4dX).

We appreciate the valuable feedback, and take this opportunity to highlight a few important enhancements we have made during the rebuttal period. Details and additional experiments can be found under individual comments to the reviewers.

**Attribute Auto-Annotation**
+ Efficiency: Reviewer aTLp request the efficiency when we apply TAD metric in datasets without attribute annotations. We response with some strategies to auto-annotate images efficiently thus mitigate the problem.
+ Sensitivity analysis: Reviewer mMtZ asks how the manually selected 25 attribute impact the performance. We response with sensitivity analysis to show the effectiveness of these attributes in task difficulty measure (Fig.2 in the pdf file). We find a more linear relationship when excludes 11 color attributes. We attribute this to the quality of obtained color attributes, because the these attributes are more susceptible to the background.
+ Evaluation of quality: Reviewer b4dX requests an evaluation of the quality of the obtained attributes. We response with experiments by annotating a small number of images ourselves. By comparing the ground-truth annotations and the results produced by CLIP model, we observe a good classification ability for CLIP model on majority attributes, while fails on some specific attributes, like "white," "pink","smooth," and "shiny".
+ Qualitive examples: Reviewer b4dX asks showing some examples of the obtained attributes, for a qualitative analysis. We response with a figure showing 6 examples (Fig.3 in the pdf file). We find the common wrong cases are color attributes. CLIP model always predicts more colors than manual annotations.

**Practicality of TAD metric**
+ Comparison with other metrics: Reviewer aTLp raised a valuable point to understand how good the TAD metric is by comparing with other metrics.
To compare different metric, we design a task selection experiment that compares the difficulty of tasks selected with highest distance.
Due to the page limit, we present some details and the corresponding table below:

  We choose three methods for comparison, which have been proposed in the few-shot learning or related area: (1) **Frechet Inception Distance (FID)** [31], (2) **Earth Mover’s Distance (EMD)** [32], (3) **Task2Vec** [ref].
  Note that those three methods rely on a pretrained model. Following [32], we use ResNet-101 pre-trained on ImageNet for them.

  ||MatchingNet|ProtoNet|RelationNet|Baseline++|APNet| Cost|
  |-|-|-|-|-|-|-|
  |FID|-3.71|-1.78|-0.57|-0.33|-3.63|8min|
  |EMD|-0.24|-1.33|1.6|-0.75|1.17| 22s|
  |Task2Vec|-0.54| -0.3|-3.64|-0.54|-2.45|100min|
  |TAD|**-8.23**|**-6.64**|**-7.39**|**-3.52**|**-7.24**| 3s|

  In the above table, we calculate the difference in accuracy between the chosen tasks and all novel tasks.
  We find that TAD significantly outperforms other three methods in identifying more challenging novel tasks across all FSL models, demonstrating the effectiveness of TAD metric.
  Furthermore, the computational efficiency of TAD greatly surpasses other methods, as illustrated in the last column of table.
  Notably, TAD requires only 3 seconds to compute across 2400 novel tasks, underscoring its advantage of ease of computation.

  [ref] Task2Vec-Task Embedding for Meta-Learning. In ICCV 2019.
+ Insight: Reviewer mMtZ encourages us to think about how our theorem and proposed metric can provide insight into few-shot learning areas.
We response with two expectations that can drive exploration within the few-shot learning domain: exploring the boundaries of FSL methods and the utilization of heterogeneous data in CD-FSL.
+ Potential applications: Reviewer b4dX request some preliminary results for potential applications of TAD metric. We response with experiments in Appendix and insights inspired by Reviewer mMtZ.

**Theory**
+ Tighter Bound: Reviewer PSzT asks a tighter bound in Theorem 1 due to the large value of VC dimension for neural networks. We response with some modifications by replace VC dimension with Rademacher complexity. Our theoretical conclusions and experimental findings will remain unchanged after the modification.
+ Infinite Sample Assumption: Theorem 2 and following analysis assume infinite labeled samples in support set, thus deviating from the few-shot learning problem. Reviewer PSzT asks whether such infinite sample assumption can be relaxed further, we response with a new condition without assuming infinite labeled samples,thereby is more suitable for few-shot learning area.

---

### Decision · Program_Chairs · 2023-09-21

**Decision:**

Accept (poster)

**Comment:**

The submission presents a metric called Task Attribute Distance (TAD) as a proxy to the difficulty of performing few-shot learning on novel tasks depending on the training tasks used to obtain the (pre)trained model. TAD is computed using predetermined image attributes, which are assumed to take on discrete and finite values. Given two learning problems, per-class attribute distributions are compared on the cross-product of the two learning problems' label spaces. TAD is then expressed as the maximum bipartite matching score between the classes in the two label spaces. For datasets with no pre-defined attributes (like mini-ImageNet), the submission proposes to use a pre-trained CLIP model in conjunction with 25 pre-defined attributes based on color, pattern, shape, and texture to annotate the datasets automatically.

Theoretical results are presented in the meta-learning setting on the relationship between TAD and the generalization error on a novel task. Empirical results are also presented on CUB, SUN, and mini-ImageNet, showing that TAD is correlated with model performance.

Reviewers note that the submission is clear (PSzT, b4dX) and tackles a central problem in few-shot learning (PSzT), and the proposed approach is an interesting and new take on defining task difficulty (aTLp, b4dX) backed by theoretical results (PSzT, aTLp, mMtZ, b4dX).

The main reviewer concerns are:

* The bound in Theorem 1 is loose, and Theorem 2 assumes that the number of samples goes to infinity, which is incompatible with few-shot learning (PSzT). The authors present an update to their theorems which replaces VC dimensions with Rademacher complexity for a tighter bound in Theorem 1 and relaxes Theorem 2's assumptions with an $\epsilon$-approximation network. This addresses Reiewer PSzT's concern.
* There is no direct comparison between TAD and related work on defining task difficulty, and the paper misses out on Task2Vec (aTLp). The authors acknowledge the Task2Vec oversight and commit to discussing it and its relationship with TAD in the final manuscript. They also present a task selection experiment comparing TAD against FID, EMD, and Task2Vec and showing TAD outperforms competing approaches. This addresses Reviewer aTLp's concern from my vantage point.
* The attribute auto-annotation approach could be prone to trial-and-error in defining the attributes, and it's unclear whether TAD is robust to different choices of 25 attributes (mMtZ, b4dX). The authors present qualitative and quantitative analyses for the auto-annotation procedure which addresses Reviewer b4dX's concern.

Following the author-reviewer discussion, the reviewer consensus is in favor of acceptance.